# Quantitative evaluation model of variable diagnosis for chest X-ray images using deep learning

Shota Nakagawa[1], Naoaki Ono[1,2]*, Yukichika Hakamata[3], Takashi Ishii[3], Akira Saito[3], Shintaro Yanagimoto[3], Shigehiko Kanaya[1,2]

1 Department of Science and Technology, Nara Institute of Science and Technology, Ikoma, Nara, Japan, 2 Data Science Center, Nara Institute of Science and Technology, Ikoma, Nara, Japan, 3 Division for Health Service Promotion, the University of Tokyo, Japan

* nono@is.naist.jp

## Abstract

The purpose of this study is to demonstrate the use of a deep learning model in quantitatively evaluating clinical findings typically subject to uncertain evaluations by physicians, using binary test results based on routine protocols. A chest X-ray is the most commonly used diagnostic tool for the detection of a wide range of diseases and is generally performed as a part of regular medical checkups. However, when it comes to findings that can be classified as within the normal range but are not considered disease-related, the thresholds of physicians' findings can vary to some extent, therefore it is necessary to define a new evaluation method and quantify it. The implementation of such methods is difficult and expensive in terms of time and labor. In this study, a total of 83,005 chest X-ray images were used to diagnose the common findings of pleural thickening and scoliosis. A novel method for quantitatively evaluating the probability that a physician would judge the images to have these findings was established. The proposed method successfully quantified the variation in physicians' findings using a deep learning model trained only on binary annotation data. It was also demonstrated that the developed method could be applied to both transfer learning using convolutional neural networks for general image analysis and a newly learned deep learning model based on vector quantization variational autoencoders with high correlations ranging from 0.89 to 0.97.

## Author summary

Machine learning for computer-aided diagnosis is not only useful for disease classification but also applicable for evaluating varying degrees of clinical characteristics. In this paper, a deep learning model was proposed that quantitatively assesses chest X-ray findings with varying annotations based on binary training data.

**Data Availability Statement:** The source code and trained models used in this study are publicly available on https://github.com/ShotaNaK-ui/chest_xray. The original diagnosis data and images

used in this study contain personally identifiable information about the subjects; therefore, they cannot be made publicly available due to ethical constraints. Requests for the transfer of anonymized data will be considered by the ethics review committee (The University of Tokyo: lifescience.adm@gs.mail.u-tokyo.ac.jp, Nara Institute of Science and Technology: k-chosei@ad. naist.jp), and interested parties should contact the corresponding author for further information.

**Funding:** This work was supported by JSPS KAKENHI Grant-in-Aid for Scientific Research (C) Grant Number 21K12111 [to NO] and Fostering Joint International Research (B) Grant Number 21KK0183 [to NO]. The funders had no role in study design, data collection, and analysis, decision to publish, or preparation of the manuscript.

**Competing interests:** The authors have declared that no competing interests exist.

## Introduction

Medical images are considered crucial for clinical diagnosis and patient management. Recent progress in computer-aided diagnosis (CAD) has improved diagnostic abilities in the clinical setting and deep learning-based techniques are accelerating the development of CAD for a wide range of medical images including X-rays, computed tomography and magnetic resonance imaging [1].

Particularly, there is a great demand for CAD in health screening programs, where radiologists or physicians need to interpret a large number of medical images. Among the various available models for CAD, deep neural networks are arising as a promising solution for detecting abnormalities, classifying image patterns, automating segmentation, and reconstituting images to increase resolution [2–5].

With the rapid advancement of deep learning techniques, their application to diagnostic support and CAD systems for identifying diseases and detecting abnormalities has become feasible. These models use various techniques, including artificial neural networks, convolutional neural networks (CNNs), and transformers, to analyze radiographic images and other clinical data. Such models have the potential to provide accurate and reliable diagnosis, leading to early detection and treatment, and thus improving patient outcomes.

In the context of diagnostic support using machine learning, it is imperative to address the issue of uncertainty assessment. Uncertainty in diagnostic support can be broadly categorized into two types: "epistemic uncertainty" and "aleatoric uncertainty" [6, 7]. Epistemic uncertainty stems from factors such as insufficient data or overly simplistic predictive models utilized in the diagnostic support process. Aleatoric uncertainty, on the other hand, arises from fluctuations in patients' symptoms or noise inherent in the observations. In machine learning, both the uncertainty inherent in the predictive model and the uncertainty inherent in the learning data themselves affect the ambiguity of predictions. In the development of diagnostic support systems, there are two primary objectives related to uncertainty assessment [8, 9]. One objective is to quantitatively characterize the statistical fluctuations inherent in the predictive models themselves, while the other is to evaluate the uncertainty associated with the labels provided by human observers, which serve as the ground truth for model training.

There have been proposed various approaches to quantify model uncertainty using deep learning [10–12]. One approach to evaluating model uncertainty is using Bayesian deep learning, which involves modeling uncertainty as probability distributions over model weights and predictions. This approach can estimate the model's uncertainty and improve the reliability of predictions, particularly in cases where the model has not been exposed to enough data during training [13]. Another approach to evaluating model uncertainty is using ensemble techniques, where multiple models are trained on different subsets of data, and their predictions are combined to produce a final output [14]. This approach can improve the accuracy of predictions and provide a measure of uncertainty based on the variability of predictions across different models. In addition, some researches have proposed uncertainty quantification techniques, such as Monte Carlo dropout and variational inference, to estimate the uncertainty associated with individual predictions [15]. These techniques can provide a measure of confidence or uncertainty associated with each prediction, allowing clinicians to make more informed decisions.

Evaluating uncertainty in the data label is more challenging in the context of machine learning [16]. From a statistical perspective of machine learning models, the evaluation of uncertainty can be regarded as a problem of assessing and replicating the probability distribution pertaining to findings and the physicians' observations [17, 18]. However, a significant

difficulty in pursuing this line of inquiry lies in the scarcity of available data, which are undoubtedly required for the quantitative reconstruction of detailed probability structures.

In this paper, we attempt to evaluate label uncertainty using CAD models based on deep learning models for diagnosis of chest X-ray images. A chest X-ray is the most commonly used diagnostic tool for detecting a wide range of pulmonary and cardiovascular diseases and is generally performed as a part of regular medical checkups. Diagnostic skills to evaluate chest X-rays vary among physicians, and many physiological deviations can appear on chest X-ray images, leading to discordant annotations [19–21]. A physician's judgment can also be viewed as a kind of probabilistic model, where the diagnosis is based on a combination of different sources of information, each with its own degree of uncertainty [22].

There are two findings which are characterized by high prevalence in health screenings; pleural thickening and scoliosis. Pleural thickening is a common finding on routine chest X-rays, and its prevalence ranges from 1.8% in teenagers to 9.8% in adults aged 60 years and older [23]. More than 90% of pulmonary thickening cases involve the apex of the lung, which is referred to as the "pulmonary apical cap" and is generally recognized as a nonspecific fibrotic change [24]. Because the apical cap possesses an irregular density at the apex and is less than 5 mm in width, its annotation can be discordant among physicians and frequently ignored when interpreting chest X-rays for health screening. However, pleural thickening can be a manifestation of respiratory diseases, such as mycobacterial infection and interstitial pneumonia [25].

Scoliosis is defined as having a lateral Cobb angle of more than 10 degrees. In previous studies, the prevalence of idiopathic scoliosis in adolescents was estimated at 1–4% [23, 26, 27]. Several models have been developed to aid in its diagnosis and monitoring [28]. For example, Horng and others proposed an automatic measurement based on the object detection method using a deep learning model and showed that the results are highly correlated ($r = 0.89$) with angles manually measured by an expert [29]. Fraiwan and others applied transfer learning and detected scoliosis with around 90% or higher accuracy [30]. Rajpurkar and others also applied a deep learning model to detect thoracic diseases and showed an accuracy of 80.6% for the detection of pleural thickening [31].

The purpose of this study is to quantify the uncertainty of the label by the probability of physicians' findings. Recent research in neuroscience, particularly in the field of cognitive psychology, has suggested that human perception and decision-making can be explained as probabilistic processes [32–34]. This has led to a growing interest in the use of probabilistic models to better understand and replicate human behavior, as well as to improve artificial intelligence and machine learning systems. For example, when identifying an object in a cluttered environment, the brain must consider various sources of information, such as visual cues, context, and prior knowledge, each with its own degree of uncertainty [35–37].

Because the output of machine learning models that predict binary determinations can mathematically be perceived as predicting the "probability of a sample belonging to a particular class." Viewing the model trained on the presence or absence of findings during a medical examination as a "probability regression model", we compared its output with the posterior probability, derived from multiple physicians' findings, demonstrating that the model indeed quantitatively evaluates depending on obviousness or difficulty of findings.

In this research, deep learning models were trained based on binary annotation from regular health checkups to predict the probability of physicians' findings. To evaluate the uncertainty of diagnosing these findings, over 80,000 chest X-ray images obtained from annual health screenings at the University of Tokyo were investigated. Our focus was directed toward the evaluation of the two prevalent findings, pleural thickening and scoliosis, owing to the following reasons: 1) these manifestations are most frequently encountered within the given

population [23], and 2) these findings are characterized by potential ambiguity and a range of severity, often leading to inter-physician variability in the resulting observations.

Using this large-scale dataset, first, the reproducibility of the classification output from deep-learning models trained using different subsets using the method of cross-validation (CV). Next, the outputs of the models were compared with the consistency of the findings of multiple physicians, in other words, the probability of finding. To verify the results, two additional physicians were asked to evaluated 1,000 images and the posterior probability of the findings based on their diagnosis. The results showed that the model provides a quantitative evaluation of the probability, which is proportional to the obviousness of findings.

To demonstrate that the proposed methodology is not dependent on the specific characteristics of any particular model, this study evaluates the approach across seven deep learning with different architectures. Six of these models are pre-trained on general photographic images, while one is a network that was randomly initialized and then trained as an autoencoder using chest X-ray images.

## Materials and methods

### Datasets

Chest X-ray images of 83,005 subjects were provided from the annual health checkup program at the University of Tokyo between 2013 and 2019. This study was approved by the Institutional Review Board of the University of Tokyo (#19-324, #21-358) and the Nara Institute of Science and Technology (#2018-M-1). An opt-out policy was announced on the website and the board waived the requirement for individual informed consent. Because this was a retrospective observational study with no therapeutic intervention, written informed consent was waived.

Chest X-rays were assessed independently by two physicians and marked as positive when one or both identified any abnormality. Abnormal findings were referred to board-certified pulmonologists for further evaluation, using available clinical information and previous X-ray images. Furthermore, 1,000 images were randomly selected from the whole image dataset and separated from the training data as the "rechecked" dataset. They were evaluated again to assess the consistency of annotation; namely, the presence of pleural thickening or scoliosis was judged by two more pulmonologists (A. S. and T. I.) in a blind manner.

### Data preprocessing

The original chest X-ray images were clipped to $3000 \times 3000$ pixels as the largest area that can be shared by all images, and resized to $256 \times 256$ pixels before being input to the model. As for data augmentation, these images were randomly shifted within 5% (13 pixels) in both the X and Y axes and rotated with a random angle between (-5, 5) degrees when used as training data. The one thousand test data were first excluded and the remaining images were randomly divided into 10 CV groups.

### Deep learning models

In this paper, our method was systematically evaluated using seven models. The seventh models were based on transfer learning using pretrained CNNs and they are fine-tuned using our chest X-ray images. The sixth model was a model using an autoencoder that was newly trained using only our datasets to demonstrate that our method can be applied to a model without using a pretrained model.

ResNet-18, ResNet-50, ResNet-101: ResNet is a family of models based on an architecture using residual blocks which minimize the residual error with reference to the input to address the problem of vanishing gradients [38]. It was the winner of the ImageNet Large Scale Visual Recognition Challenge 2015 (ILSVRC2015) [39]. In this paper, three pre-trained models on the ImageNet database, ResNet-18, ResNet-50, and ResNet-101, were used, which are composed of corresponding depths, 18 layers, 50 layers, 101 layers, and the number of parameters were 12 million, 26 million and 45 million, respectively.

DenseNet-201: DenseNet introduces connections between several preceding layers which improve feature propagation/reuse and drastically reduce the number of parameters [40]. It makes shortcuts to the connections between the layers close to the input/output. DenseNet-201, as the name suggests, is composed of 201 layers deep with 20 million parameters. It was trained on the ImageNet database.

EfficientNet-b0: EfficientNet is a set of models provided by the optimized scaling methods that expand the depth and width of the networks. EfficientNet-b0 is the baseline model which is optimized by the AutoML methods and whose performance is comparable with the preceding models, though the number of parameters is much smaller (about 4 million) [41]. The model was trained using the ImageNet database.

ConvNeXt-B: ConvNeXt is a "modernized" architecture of a standard ConvNet (ResNet) model toward the design of a hierarchical vision transformer [42]. It achieved the same level of performance with transformer-based state-of-the-art models without introducing any attention-based modules. ConvNeXt-B, with 89M parameters, was trained on the ImageNet database.

VQ-VAE: Vector Quantized Variational Auto Encoder (VQ-VAE) [43] is a model to effectively extract features even starting from scratch. In VQ-VAE, first, pixel intensities are input into a CNN to obtain a map of feature vectors. Then, each feature vector is quantized by replacing it with the closest vector in a limited set of vectors called a "code book". A decoder is defined as a set of deconvolution layers that reconstruct input images from the quantized feature vectors. In this study, the feature extraction network was trained as an unsupervised auto-encoder to reconstruct the whole chest X-ray images of our dataset, using mean squared error as the loss function.

These networks were concatenated with classifiers with dense layers. All features were flattened and input into the classifier based on fully connected networks. The latter is a multi-target classifier to predict the diagnosis of both pleural thickening and scoliosis. The loss of the classifier $l_{Cls}$ is defined as the sum of the binary cross-entropy with logit for both classifiers, pleural thickening and scoliosis. Then the whole model was trained as a multi-target classifier for fine-tuning. In the case of the autoencoder-based model, the whole network was trained using the sum of the reconstruction error and classification error as a loss function. Fig 1 shows the architectures of the model.

## Training and validation

According to the same procedure as 10-fold CV, the training dataset was divided to 10 subsets, and a model was trained using nine-tenths of the samples and validated using the excluded one-tenth of the samples (Fig 2). The adaptive moment estimation (ADAM) [44] was used for optimization and the initial learning rate was set to $10^{-4}$. After 5 epochs of learning, the validation accuracy was almost saturated in almost all cases so training was halted. This procedure was repeated 10 times so the results of validation data in this manuscript are the average of the whole sample using 10 models that were trained independently starting from scratch.

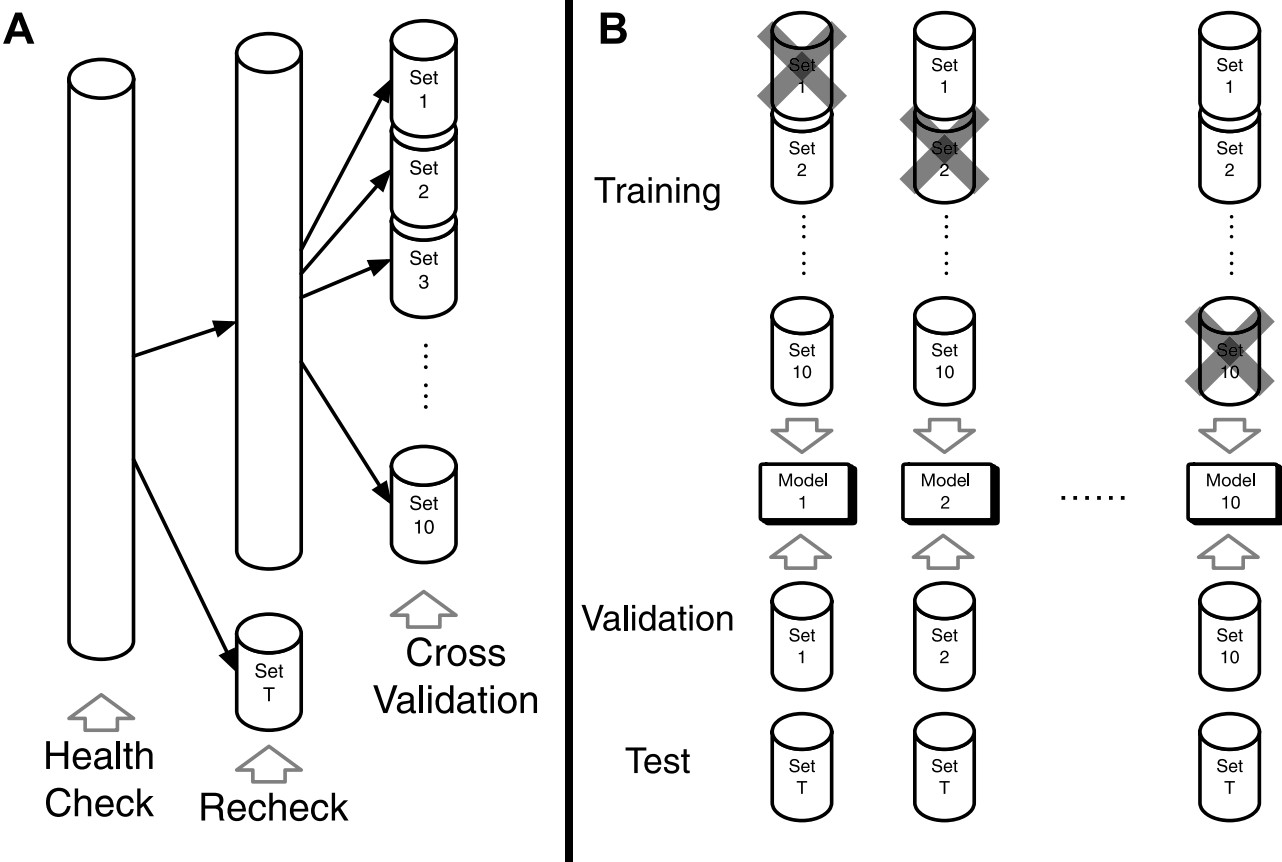

**Fig 1. The architecture of the networks.** (A) Transfer learning. The output from the CNN models (ResNet, DenseNet, EfficientNet, ConvNeXt) pre-trained using the ImageNet dataset is regarded as feature vectors and connected to a classifier with dense networks. (B) Classifier with a pre-trained autoencoder. The network based on VQ-VAE was trained as an autoencoder using chest X-ray images. Then, extracted features were connected to a classifier with convolution layers and a dense network. In all cases, the whole network was trained (fine-tuned) for the training datasets.

Finally, the one thousand rechecked data were evaluated using these 10 models, which were independently trained by the 10-fold CV procedure. Fig 2 presents the scheme of validation and testing.

### Evaluation of posterior probability

It was assumed that diagnosis by physicians follows a Bernoulli process with a probability $p$. Because $p$ is unknown, a Bayesian model was considered to evaluate the expected probability of the findings. Note that the original diagnosis represents the union of the findings by two physicians, therefore, the probability of a positive finding P and a no findings N is $p^2 + 2p(1 - p)$ and $(1 - p)^2$, respectively. By contrast, the verification of the rechecked dataset by two physicians was counted independently, and the findings can be classified as {PP, PN, NN}. According to the results of two stages of diagnosis, the one thousand test data can be categorized into six groups, $D \in \{(P, PP), (P, PN), (P, NN), (N, PP), (N, PN), (P, NN)\}$.

Taking both first and second evaluations into account, the posterior probability is given using Bayes theorem. Because there is no preliminary information, it can be simply assumed that the prior $p_0(x) \equiv 1$. For example, when the first diagnosis and both physicians in the

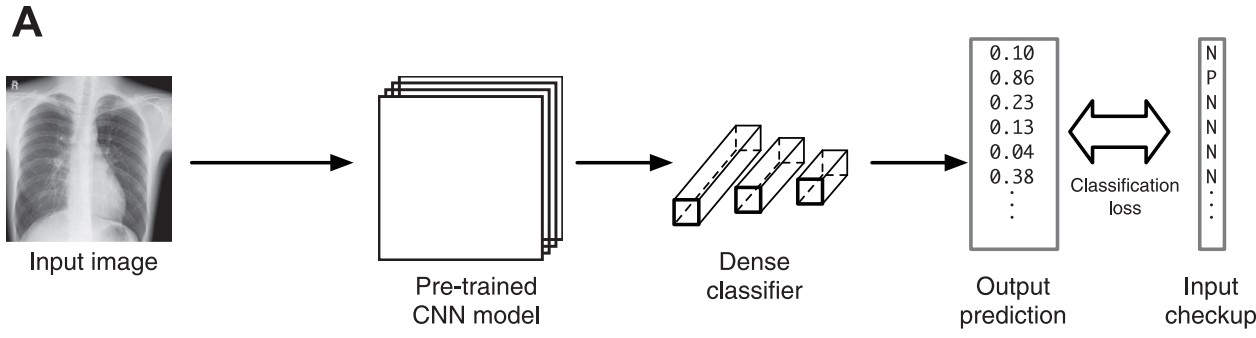

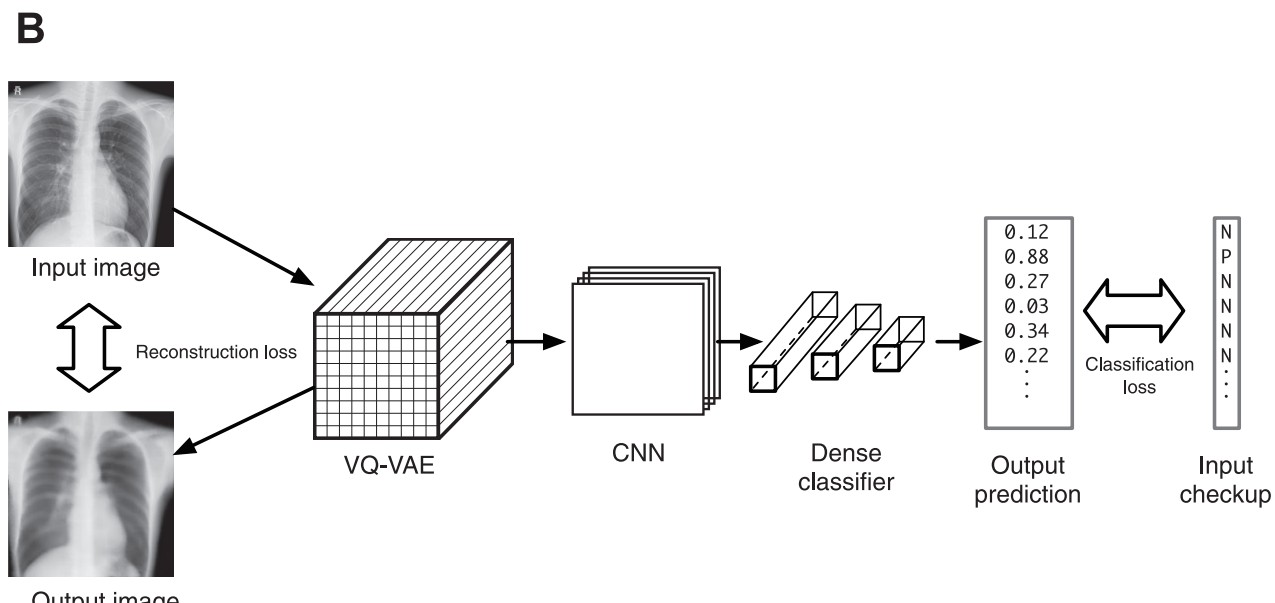

**Fig 2. A schematic illustration of the datasets of the validation scheme.** (A) One thousand subjects were excluded from the test dataset. The remaining data were split into 10 cross-validation (CV) datasets. (B) One of the CV datasets was excluded as validation data and the de novo networks were trained from scratch, which were evaluated using the validation data and the one thousand rechecked data.

second evaluation all provide positive results, the posterior probability is given as follows;

$$
\begin{aligned}
p(x|D) &= \frac{p(D|x)p_0(x)}{\int p(D|x)p_0(x)} \\
p(x|\mathrm{P}, \mathrm{PP}) &= \frac{(x^2 + 2x(1-x)) \times x^2}{\int_0^1 (x^2 + 2x(1-x)) \times x^2},
\end{aligned}
\tag{1}
$$

where $p_0(x)$ denotes the prior probability.

The expected value of this posterior distribution is,

$$
\hat{p}(\mathrm{P}, \mathrm{PP}) = \langle p(x|\mathrm{P}, \mathrm{PP}) \rangle = \int_0^1 x p(x|\mathrm{P}, \mathrm{PP}) = \frac{10}{3}\frac{7}{30} = \frac{7}{9}.
$$

According to the physicians' findings, all possible combinations of the first and the second diagnosis can be classified into six cases, and the expected probability can be computed in the same way. The expected averages of posterior probabilities are as follows, $(\hat{p}(P, PP), \hat{p}(P, PN), \hat{p}(P, NN), \hat{p}(N, PP), \hat{p}(N, PN), \hat{p}(N, NN)) = (\frac{7}{9}, \frac{4}{7}, \frac{3}{8}, \frac{1}{2}, \frac{1}{3}, \frac{1}{6})$.

The sigmoid function $x(u) = \frac{1}{1+e^{-u}}$ was applied to the output of the network so that the model was trained using binary cross-entropy as a loss function. This implies that the outputs ($u$) of the model are the logit values of the probability ($x$) of the classification and the training minimizes the difference between the predicted probability and that of the targets. However, the actual outputs would depend on how closely the model is optimized to the ideal probability distribution. Therefore, to take the difficulty and complexity of the problem into account, a coefficient $\lambda$ that scales the output $u$. which expresses the logit of the probability, was introduced.

Consider the distribution of the scaled probability,

$$u_i^\star = -\lambda u_i \tag{2}$$

$$p_i^\star = \frac{1}{1 + e^{-u_i^\star}}. \tag{3}$$

In addition, consider the transformation of the distribution of the posterior probability from $x$ to its logit $u$. For example, Eq 1 can be transformed as,

$$p_x(x|P, PP) = \frac{10}{3}(x^4 + 2x^3(1 - x)) \tag{4}$$

$$p_u(u|P, PP) = p_x(x(u)) \times \frac{dx}{du}(u) \tag{5}$$

$$= \frac{10}{3}\frac{1 + 2e^{-u}}{(1 + e^{-u})^4}\frac{e^{-u}}{(1 + e^{-u})^2}. \tag{6}$$

Two scale parameter $\lambda_s$ and $\lambda_p$ were introduced for each finding pleural thickening, and scoliosis, and optimized to minimize the average of the Euclidean distance between the distribution of $u_i^\star|D$ and the posterior distribution $p_u(u|D)$ in all six cases.

The rechecked datasets were grouped according to six cases of physicians' findings in the first and second evaluations, and the average output of the model was determined for each group. To evaluate the agreement between the predictions made by the model and the probability of a diagnosis made by a physician, the correlation coefficient between the average and the expected value of the posterior probability was calculated.

## Saliency map

As a method for explaining the results of analysis, a saliency map method that visualizes which areas of an image contributed to the judgment is widely used. In this study, we used the Grad-CAM method [45] to highlight the areas that affected the findings of pleural thickening and scoliosis, and physicians confirmed whether they were appropriate.

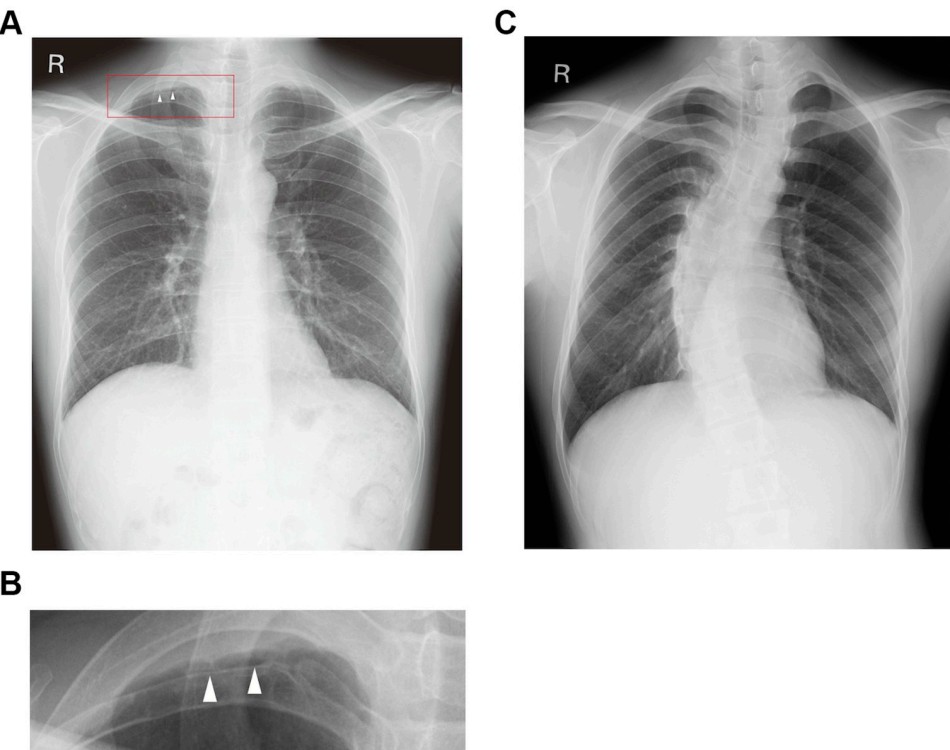

**Fig 3. Representative chest X-ray images.** (A) An example of pleural thickening. (B) A magnified image of the right apex with pleural thickening (area marked by a red rectangle), with white triangles indicating pleural thickening. (C) An example of scoliosis.

## Results

### Data statistics

Among the 83,005 chest X-rays obtained from the annual health examinations between 2013 and 2019, the gender rate of female/male was 36%/63%, and ages ranged from 18 to 81 years with a mean and SD of 30.6 ± 11.0. In total, 16,196 abnormal findings were observed on 9,314 chest X-rays by at least one physician. Pleural thickening was the most common finding (4.46%; $n = 3,825$), while scoliosis was the second most frequently annotated (1.58%; $n = 1,309$). Fig 3 shows an example of the images of these findings.

Next, 1,000 randomly sampled "rechecked" data were reviewed to evaluate the consistency of the findings. Fig 4 shows the agreements of the initial health check and recheck by the two physicians. The increase in the findings at later checkouts implies that these physicians read images more carefully in the recheck process. The Cohen's $\kappa$ coefficient [46] between the findings by the two physicians were 0.60 for pleural thickening and 0.79 for scoliosis, suggesting a high level of agreement, and hence, inter-variation could be ignored in this study.

### Model evaluation

Six classifier models were constructed using both transfer learning based on pre-trained CNNs and a de novo model based on an autoencoder.

In the cases of transfer learning, "ResNet-50", "ResNet-101", "DenseNet-201", "Efficient-Net-b0", and "ConvNeXt" were used and a classifier was appended to each model. In the case

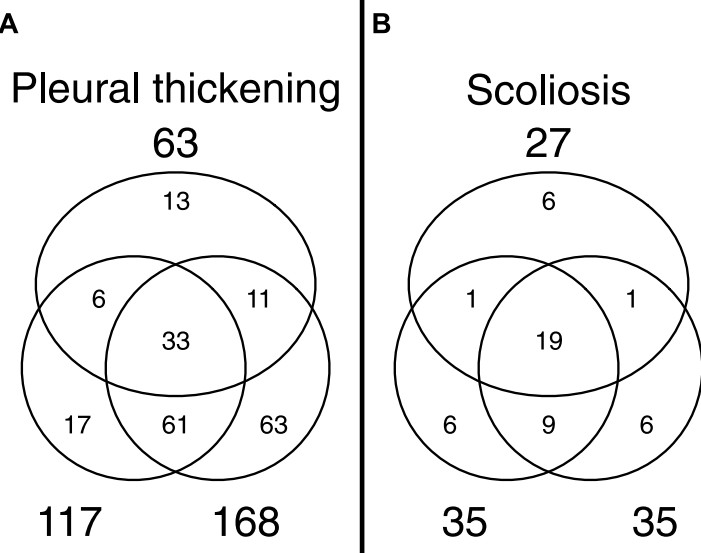

**Fig 4. Venn diagram of the agreements of findings.** Upper circles represent the findings in the initial health check. Lower circles represent those of the recheck. (A) Pleural thickening. (B) Scoliosis.

of the de novo model, an autoencoder based on the VQ-VAE was used. After training the auto-encoder, the model was concatenated to the classifier network, and the extracted latent vectors were inputted into the classifier.

Tables 1 and 2 summarize the averaged performance of the 10 CV models, namely, area under the ROC curve (AUC), F1 score, precision (Prec.), recall (Rec.), specificity (Spec.), and accuracy (Acc.). The upper six rows present models based on transfer learning and the last row presents the autoencoder trained from scratch using chest X-ray images.

## Evaluation of uncertainty

Next, the output of these models using the rechecked dataset was evaluated. The output using the same input images was consistent among the different CV models, though they were not identical due to the randomness in the training process. Fig 5 shows examples of scatter plots that compare the outputs of the different CV models based on EfficientNet. The results showed that the output of the prediction model for the same sample can vary widely depending on the randomness. The correlation of the output between all 45 pairs of different CV models and

**Table 1. Comparison of the models for pleural thickening.**

|  | AUC | F1 | Prec. | Rec. | Spec. | Acc. |
|---|---|---|---|---|---|---|
| **ResNet-18** | 0.941 | 0.916 | **0.993** | 0.850 | **0.894** | 0.852 |
| **ResNet-50** | 0.932 | 0.906 | **0.993** | 0.832 | 0.881 | 0.834 |
| **ResNet-101** | 0.899 | 0.899 | 0.990 | 0.823 | 0.827 | 0.824 |
| **DenseNet-201** | 0.901 | 0.905 | 0.990 | 0.833 | 0.821 | 0.833 |
| **EfficientNet-b0** | **0.943** | **0.931** | **0.993** | **0.876** | 0.875 | **0.876** |
| **ConvNext-B** | 0.899 | 0.892 | 0.991 | 0.812 | 0.840 | 0.813 |
| **VQ-VAE** | 0.823 | 0.850 | 0.985 | 0.749 | 0.755 | 0.749 |

**Table 2. Comparison of the models for scoliosis.**

|  | AUC | F1 | Prec. | Rec. | Spec. | Acc. |
|---|---|---|---|---|---|---|
| ResNet-18 | **0.994** | 0.984 | **0.999** | 0.969 | **0.967** | 0.969 |
| ResNet-50 | 0.993 | 0.985 | **0.999** | 0.971 | 0.959 | 0.971 |
| ResNet-101 | 0.968 | 0.966 | **0.999** | 0.936 | 0.915 | 0.935 |
| DenseNet-201 | 0.974 | 0.905 | **0.999** | 0.935 | 0.914 | 0.935 |
| EfficientNet-b0 | **0.994** | **0.986** | **0.999** | **0.972** | **0.967** | **0.972** |
| ConvNext-B | 0.970 | 0.967 | **0.999** | 0.937 | 0.925 | 0.937 |
| VQ-VAE | 0.958 | 0.959 | 0.998 | 0.922 | 0.892 | 0.922 |

their averages and standard deviation were 0.906±0.016 in the cases of pleural thickening, and 0.95±0.016 in the cases of scoliosis, respectively. More specifically, the correlation depended on the group of diagnostic results (Table 3). The reason why the correlation is low for images in which no findings were observed is considered to be that the distribution is saturated because the output values are fixed around 0. It can be inferred that the prediction of the model reflects the findings very well, but the output can vary stochastically in a considerable range depending on the learning process and datasets. Moreover, the results indicated that the output values of the prediction models were statistically consistent with the physician's findings.

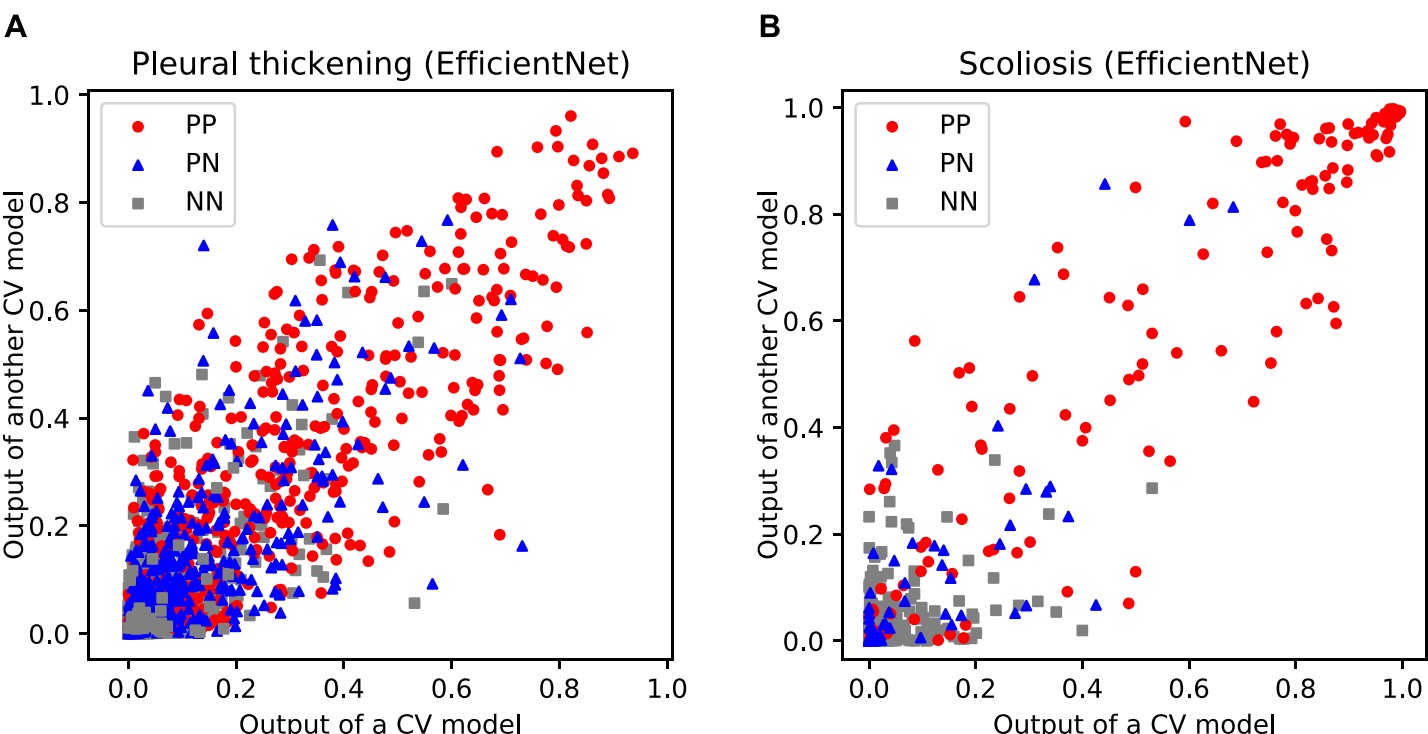

**Fig 5. Reproducibility of the outputs.** Comparison of output of the different CV models. (A) Pleural thickening. (B) Scoliosis. The red circles represent the samples judged to be positive by both physicians. The blue triangles represent the samples judged to be positive by either one. The gray squares represent samples judged to be negative by both.

**Table 3. Correlation of the outputs.**

|  | Pleural thickening | Scoliosis |
|---|---|---|
| PP | 0.882±0.024 | 0.907±0.034 |
| PN | 0.795±0.044 | 0.792±0.091 |
| NN | 0.747±0.041 | 0.499±0.111 |

To understand the variety of predictions based on a statistical model, the distribution of the output of the classifiers was compared with the posterior distribution given as the Bayesian model. The scaling coefficients λ described in Eq 2 by minimizing the mean squared error between output and expected distributions, and $\lambda_p = 0.79$ for pleural thickening and $\lambda_s = 0.22$ for scoliosis were obtained. These scaling coefficients seems to be corresponding to the AUC of the classifiaion results; i.e., in the case of scoliosis, features were more easily observed so that the differences in the output values for individual images were much larger than that of pleural thickening.

Figs 6 and 7 show the comparison of the distribution of the classifier model and the posterior probability using all models, namely, the distribution of $p(u^\star|D)$. Though the variances of the output of the model in each case were rather large, their averages were highly correlated with the posterior probability of the findings evaluated by the Bayesian model. Table 4 show the correlation coefficients of the expected value and average of the output.

From the acquired results it can be argued that the distribution of the outputs of the classifier is consistent with the Bayesian model that regards the physician's findings as a

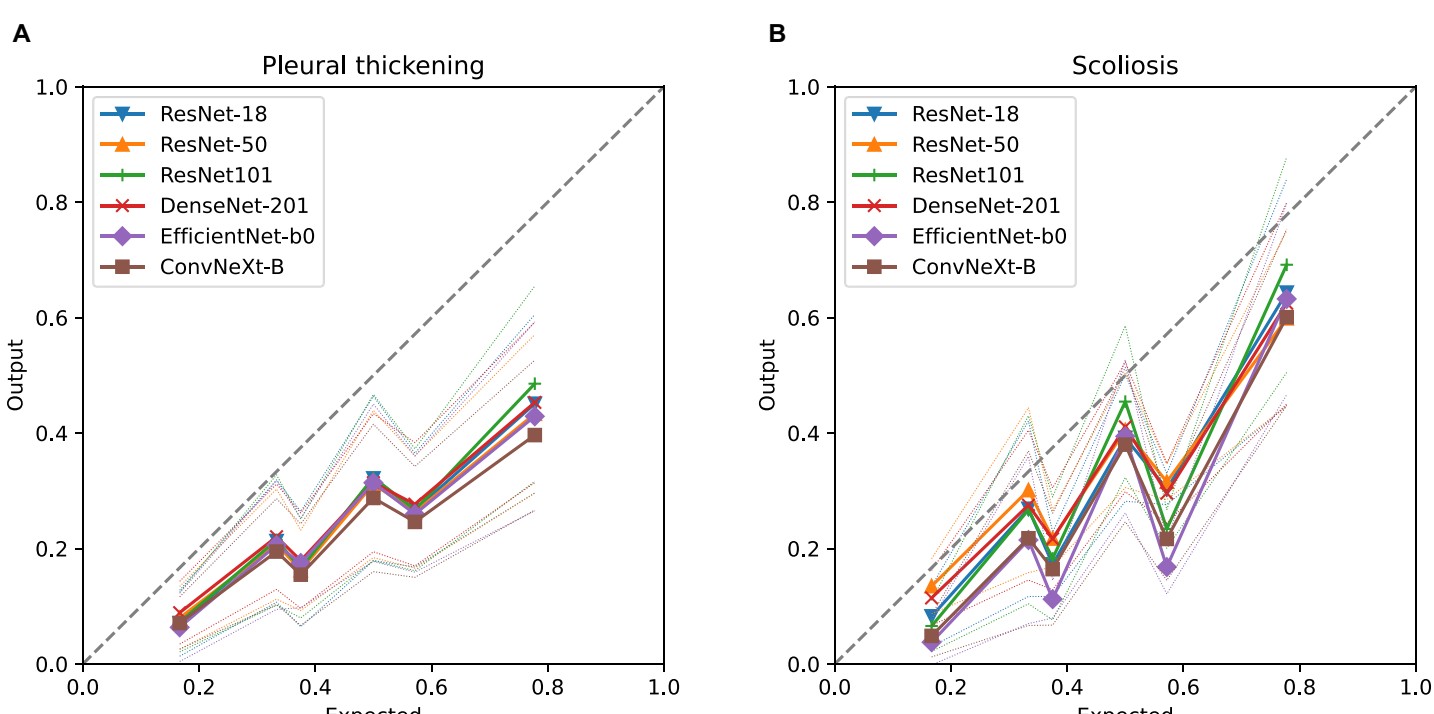

**Fig 6. Comparison between the predicted probability using pretraiend models and physicians' findings.** The x-axis represents the expected value of the posterior distribution. The y-axis shows the distribution of the prediction. The marked line represents the average of the image output values corresponding to the combination of physician findings, and the dotted line represents their standard deviation.

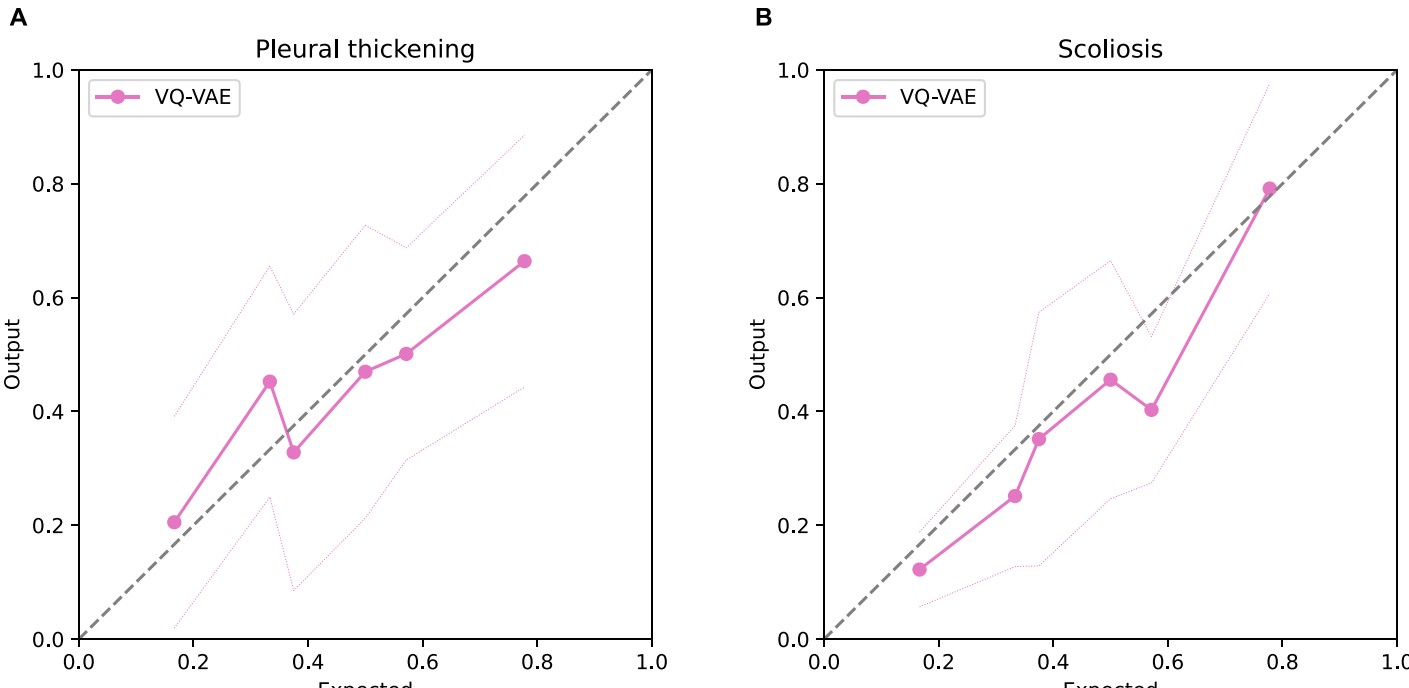

**Fig 7. Comparison between the predicted probability using newly trained models and physicians' findings.** The x-axis represents the expected value of the posterior distribution. The y-axis shows the distribution of the prediction. The marked line represents the average of the image output values corresponding to the combination of physician findings, and the dotted line represents their standard deviation.

probabilistic process, either using transfer learning trained by the ImageNet dataset or using an autoencoder trained from scratch with chest X-ray images.

## Explainability

Besides the correlation of the output with the probability of the findings, the saliency maps of each finding using the Grad-CAM method were investigated. Fig 8 visualizes the heatmaps of the gradients, i.e., areas where the features of the radiographic images positively contributed to increasing the probability of the findings. It is clear that the classifier mainly focused on the lung apex in the case of pleural thickening, and the curvature of the spine in the cases of scoliosis, which are reasonable, from the viewpoint of physicians. Similar visualization was obtained using other models.

**Table 4. Correlation with physicians' findings.**

|  | **Pleural thickening** | **Scoliosis** |
|---|---|---|
| ResNet-18 | 0.96 | 0.95 |
| ResNet-50 | 0.96 | 0.92 |
| ResNet-101 | 0.97 | 0.90 |
| DenseNet | 0.96 | 0.93 |
| EfficentNet | 0.96 | 0.89 |
| ConvNeXt | 0.96 | 0.92 |
| VQ-VAE | 0.94 | 0.96 |

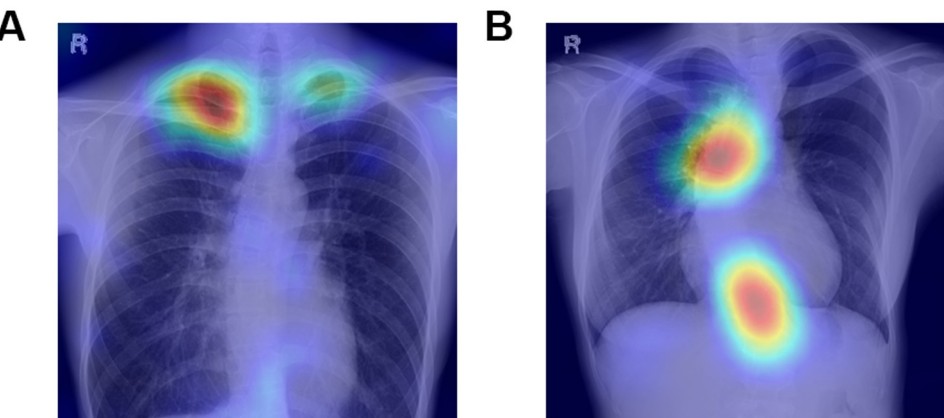

**Fig 8. Saliency maps for pleural thickening and scoliosis using the EfficientNet model.** (A) Pleural thickening. (B) Scoliosis.

## Discussion

Here we introduced deep learning models to quantitatively evaluate the obviousness of variable diagnosis as the probability of physicians' findings for pleural thickening and scoliosis. It was shown that correlation between predicted output was correlated with the expected posterior probability using several different models of deep learning with and without pre-training. This indicates that our approach to evaluate the posterior probability of the findings can be generally applicable as a quantitative measure of the significance of findings. As demonstrated in this research, machine learning models are generally designed to reproduce the probability distribution of sample classification based on input data. For CAD applications, it would be more helpful to present such quantitative indicators and leave the final judgment to the annotators.

In this regard, our model would be suitable as an approach for CAD systems, particularly for the tasks of interpreting health screening chest X-rays. The strength of our study is that we could utilize a large dataset of chest X-ray images reviewed by two experienced physicians and further supervised by pulmonologists. Detailed annotations of a large dataset enabled us to establish a prediction model with high quality. On the other hand, the data set used for the analysis was derived from Japanese participants and more than half were aged 20 to 30 years, so it will be necessary to analyze them using a wider range of datasets, in order to generalize the performance of prediction.

The classification of pleural thickening seems much more difficult than that of scoliosis. One of the reasons for the low accuracy in the detection of pleural thickening can be attributed to the low resolution of the input images. Though physicians focus on detailed textures in magnified high-resolution images to investigate pleural thickening, the input images used in this study were rescaled to a much lower resolution. A more sophisticated model, such as multi-resolution CNN will be needed to obtain a clearer classification of pleural thickening. Another possible explanation is that the physicians' findings themselves have a larger amount of uncertainty in the case of pleural thickening. By combining and comparing our proposed method with other methods for measuring model uncertainty, it would be possible to quantify the inherent uncertainties that cannot be removed even using more complex models.

According to the performance comparison of transfer learning models listed in Tables 1 and 2, models with a large number of parameters do not necessarily give better results. Similar

trends can be seen in other previous studies using transfer learning [30]. Unlike learning using a large number of image samples, in fine tuning performed using a limited number of samples, it is thought that learning efficiency will decrease if the number of parameters is too large. In particular, when performing transfer learning on images that are in a different domain from general photographs, such as medical images, care must be taken to use a smaller model to the extent that sufficient performance can be obtained.

The proposed model showed high accuracy, precision, and AUC for detecting scoliosis. It was noteworthy that Grad-CAM images showed the area around the lung apex in the case of pleural thickening, and the curvature of the spine in the cases of scoliosis. A CAD model to quantitatively evaluate the saliency of representative chest X-ray annotations, pleural thickening and scoliosis, through classifiers based on deep learning seemed feasible and clinically relevant.

Note that probabilistic evaluation proposed in this approach is not suitable for determining findings that are directly linked to critical diagnosis such as pneumonia, lung cancer, etc. The proposed method primarily focuses on findings with inherently uncertain annotations, such as pleural thickening and scoliosis. Providing a quantitative evaluation of such symptoms in the form of a probability distribution will support physician's judgment in diagnosis.

Even though the binary annotation was available as the training standard, the proposed method provided the outputs which were strongly correlated with the probability of annotations by physicians. This outcome indicates that the outputs of the neural network can be regarded as a quantitative index for probabilistic reasoning in diagnosis [47] and will help to objectify the standards of diagnosis and achieve agreement among annotators. Our approach paves the way for developing on-site utilities for routine and screening chest X-ray images.

## Author Contributions

**Conceptualization:** Naoaki Ono.

**Data curation:** Yukichika Hakamata, Takashi Ishii, Akira Saito.

**Formal analysis:** Naoaki Ono.

**Funding acquisition:** Shigehiko Kanaya.

**Investigation:** Shota Nakagawa, Akira Saito.

**Methodology:** Naoaki Ono.

**Project administration:** Shintaro Yanagimoto.

**Resources:** Yukichika Hakamata, Takashi Ishii, Akira Saito.

**Software:** Shota Nakagawa.

**Supervision:** Naoaki Ono, Shintaro Yanagimoto, Shigehiko Kanaya.

**Validation:** Takashi Ishii, Akira Saito.

**Visualization:** Shota Nakagawa.

**Writing – original draft:** Naoaki Ono, Akira Saito.

**Writing – review & editing:** Naoaki Ono, Akira Saito.

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
