## [Decision Letter · Decision Letter 0]

17 Aug 2023

PDIG-D-23-00283

Quantitative evaluation model of variable diagnosis for chest X-ray images using deep learning

PLOS Digital Health

Dear Dr. ONO,

Thank you for submitting your manuscript to PLOS Digital Health. After careful consideration, we feel that it has merit but does not fully meet PLOS Digital Health's publication criteria as it currently stands. Therefore, we invite you to submit a revised version of the manuscript that addresses the points raised during the review process.

Please submit your revised manuscript within 60 days Oct 16 2023 11:59PM. If you will need more time than this to complete your revisions, please reply to this message or contact the journal office at digitalhealth@plos.org. Please include the following items when submitting your revised manuscript:

We look forward to receiving your revised manuscript.

Kind regards,

Sivaramakrishnan Rajaraman, Ph.D.

Academic Editor

PLOS Digital Health

Journal Requirements:

1. We ask that a manuscript source file is provided at Revision. Please upload your manuscript file as a .doc, .docx, .rtf or .tex.

2. Please provide separate figure files in .tif or .eps format only and remove any figures embedded in your manuscript file. Please also ensure that all files are under our size limit of 10MB.

Additional Editor Comments (if provided):

I suggest that the authors answer all the queries from the reviewers and improve clarity and readability of the manuscript.

Reviewers' comments:

Reviewer's Responses to Questions

**Comments to the Author**

1. Does this manuscript meet PLOS Digital Health’s publication criteria? Is the manuscript technically sound, and do the data support the conclusions? The manuscript must describe methodologically and ethically rigorous research with conclusions that are appropriately drawn based on the data presented.

Reviewer #1: Partly

Reviewer #2: Partly

Reviewer #3: Partly

Reviewer #4: Yes

2. Has the statistical analysis been performed appropriately and rigorously?

Reviewer #1: Yes

Reviewer #2: I don't know

Reviewer #3: Yes

Reviewer #4: Yes

3. Have the authors made all data underlying the findings in their manuscript fully available (please refer to the Data Availability Statement at the start of the manuscript PDF file)?

Reviewer #1: No

Reviewer #2: Yes

Reviewer #3: No

Reviewer #4: Yes

4. Is the manuscript presented in an intelligible fashion and written in standard English?

Reviewer #1: Yes

Reviewer #2: Yes

Reviewer #3: Yes

Reviewer #4: Yes

5. Review Comments to the Author

Reviewer #1: This study presents a deep-learning model that quantitatively assesses chest X-ray findings with inter-clinician varying annotations. The authors implemented five different deep-learning models from 4-8 years ago to detect pleural thickening (PT) and scoliosis (S) in chest X-ray scans. For their analysis, they used an X-ray dataset containing over 82k chest X-ray scans labeled positive or negative with respect to PT and S. Two annotators re-examined 1k of the scans (independently). While I definitely understand the potential of such a study as a starting point for AI-based PT and S diagnosis, the manuscript feels a bit lacking with a “take-home message” as I’ll explain below. The manuscript is written and detailed very well and (almost) whenever I looked for some piece of information it was in the place I expected it to be so I could easily navigate. IMHO, this manuscript should be accepted upon bottom-line sharpening and better benchmarking.

What exactly is the final model suggested by the authors? It is unclear. Is it the (highest-scoring) efficient-net-based model? Why is the VAE so highlighted (across paper and in the abstract as well)? It underperformed all other models in almost every metric. Maybe I misunderstood something so it should be further elaborated… 

Why did the authors use ResNet 50 and 101? Why not start with the simplest (i.e., 18)? It’s apparent that the ResNet-50 outperformed both 101 and the DenseNet-201 (and the VAE). Actually, I am not sure I was convinced that it’s not on-par with the efficient-net for the scoliosis learning task (any confidence intervals/significance evaluation for the difference? See [1] for example). I would have tried first a simpler approach to make sure it’s not an overkill.a

Benchmarking? Looks like the authors compared their models to none of the currently available methods, either generic, such as ConvNeXt [2] or medical-oriented such as REMEDIS [3], CheXzero [4], etc… It doesn’t have to be these ones but showing superiority against at least one state-of-the-art method is the minimum, o.w., what’s the point of the “new model”?

The evaluation of research reproducibility poses challenges due to the absence of evaluation code or fine tuned weights for baselines or method itself (GitHub link is broken). Also, the exact hyperparams used and the optimization methods were not detailed in the manuscript.

What was the positives prevalence in the full dataset? Test set? Did you do a stratified CV? If so, you should have had around 10 positive cases (out of 1000) in the test set which makes it very hard to provide a meaningful evaluation. Did you try data augmentation?

Minor comments:

Add ref “In previous studies, the prevalence of idiopathic scoliosis in adolescents was estimated at 1 to 4%”

“nine-tenth” -> “nine tenths”

“annottation”

“is by” ?

Fig. 2

“Dence“

Fig. 3

I would add a boundary box to refer to the zoomed area 

Fig. 5

The correlation of PP is qualitatively apparent (and expected) but for PN and NN it’s unclear. I would expect higher correlation for the NNs (compared to PNs) assuming that if both annotators reached the same conclusion (an in PP), why would the models not? It’s hard to compare without numbers/zoom-in to the relevant region. I suggest plotting each label in a different plot and providing the R^2.

Fig. 6

“λp and λp” -> “λp and λs”

What were the (separate) mean R^2s across all 45 pairs of models for PP PN NN?

Not sure I understand this sentence: “scoliosis was more clearly separated in the feature space so that the variance in output values was much larger than that of pleural thickening.” shouldn’t it be the other way around? If it’s easily separable the model’s variance should be lower…

“Tables 1 and 2 summarize the performance of the models” -> “Tables 1 and 2 summarize the mean performance of the 10 cross-validated models”

AUC is defined for the first time in the discussion but used a few times before that.

What’s the re-checkout? Please define new terminology you use.

Feels like it should go to the intro and be more concise in the discussion (1-2 sentences): “Pleural thickening is a common finding on routine chest X-rays and its prevalence ranges from 1.8% in teenagers to 9.8% in adults aged 60 years and older. More than 90% of pulmonary thickening cases involve the apex of the lung, which is referred to as the ‘pulmonary apical cap’ and is generally recognized as a non-specific fibrotic change. Because the apical cap is an irregular density at the apex and is less than 5 mm in width, its annotation can be discordant among physicians and frequently ignored when interpreting chest X-rays for health screening. However, pulmonary thickening can be a manifestation of respiratory diseases such as mycobacterial infection and interstitial pneumonia”

P.s., please add line numbers in future submissions o.w. it makes commenting and following-up very inconvenient.

[1] SLIViT: a general AI framework for clinical-feature diagnosis from limited 3D biomedical-imaging data https://doi.org/10.21203/rs.3.rs-3044914/v1

[2] A ConvNet for the 2020s https://doi.org/10.48550/arXiv.2201.03545

[3] Robust and data-efficient generalization of self-supervised machine learning for diagnostic imaging https://doi.org/10.1038/s41551-023-01049-7

[4] Expert-level detection of pathologies from unannotated chest X-ray images via self-supervised learning https://doi.org/10.1038/s41551-022-00936-9

Reviewer #2: I want to express my sincere appreciation for the opportunity to review the manuscript titled “Quantitative evaluation model of variable diagnosis for chest X-ray images using deep learning”. Having carefully examined the research, I would like to convey my interest and appreciation for the results presented in this study. The authors' work in shedding light on uncertainty of the deep learning models in diagnosis of chest X-ray images. The methodology employed was sound, and the analysis of the data yielded insightful conclusions. I would like to acknowledge the researchers for their effort. While the manuscript is indeed promising, I would like to offer a few notes for potential improvement. They are divided in major and minor notes: 

Major

1. Authors mentioned in the discussion developing several models with different architectures followed by evaluating the posterior probability of the findings. However, within the limitations of my understanding, they only evaluated the VQ-VAE model.

2. What is the benefit from building multitarget model compared to separate model for each diagnosis.

Minor

1. It would be helpful to mention the other transfer learning models developed in the abstract as the best model was one of them. More effort in adding details about the method and results should be done to provide readers with more details about the methodology used and the promising results.

2. More citations must be included in key sentences in the introduction and discussion.

3. Authors mentioned in the introduction that pleural thickening and scoliosis to be the most common findings in chest x-ray datasets followed by prevalence of the two findings. It’s recommended to added references that supports the first claim and add the references [21] and [35] to the paragraph as the pleural thickening and scoliosis prevalence was taken from them. They weren’t cited before this sentence.

4. In the introduction the pleural thickening and scoliosis were both mentioned to be the most prevalent in the population while reference [21] used only supports pleural thickening.

5. In the introduction try to mention the objective of the study in the last paragraph. It is advised that any justification for your objectives is mentioned before the objectives. Moreover, the objectives of the authors study should contain less of the methodology.

6. According to ethical considerations, authors should provide details about the IRB ethical approval number.

7. In the results the authors reported the average of correlation coefficient for the 45 pairs of models. It would be more informative to report the standard deviation to acknowledge the distribution of the values.

8. The interpretation for lower accuracy in detecting pleural thickening is duplicated in the results and discussion. A more scholarly approach would involve addressing this matter exclusively within the discussion segment, coupled with the recommend solutions. 

9. More on table 1 and 2 results should be presented and discussed in the results and discussion sections. How different models architectures contributed to their performance? Is there any studies that used them and have a matching results?

10. Comparison and discussion of the findings in this research with other studies is advisable. It would be interesting how the results of this study can be compared to other AI models developed for diagnosis of X-ray images. 

11. Language editing for the manuscript would provide clearer presentation for the readers. 

In conclusion, I believe that this manuscript has the potential to make a valuable contribution to the scientific community. The authors' commitment to rigorous research and their ability to generate thought-provoking results are truly commendable. I am looking forward to seeing how the authors address the points raised and further refine their work.

Reviewer #3: 1. Mention the achieved accuracy in the abstraction.

2. The previous research work is mentioned in the introduction section please create a separate section with the heading Literature Survey.

3. Add more literature citations in support of the literature survey.

4. Cross-verify the figure citation (It is not Figure 1 replace it with Figure 1.)

5. Label Figure 3 as Figure 3a, 3b, and 3c instead of label upper and lower.

6. State the limitation of your model.

7. The data collection process is explained in a good style of writing.

Reviewer #4: Quantitative evaluation model of variable diagnosis for chest X-ray images using deep learning

The author(s) proposed a deep learning-based approach using vector-quantized

variational autoencoder to quantitative assess the probability

of detecting pleural thickening and scoliosis.

The author(s) “assumed the annotator's binary findings, that is,

the presence or absence depends on the degree of radiological changes, and evaluated the probability of making an annotation.” What about intra-inter-observer variation, which has been well-established in the literature? For example, one annotator can make different findings depending on how they feel on the day. Also, two annotators can come up with two findings given the same X-ray image.

The researcher(s) used two physicians to independently assess the Chest X-rays. How does this account for inter-intra-observer variability?

One of the biggest challenges with X-ray is that it is 2D, thus only giving one projection of the organ being imaged. As a result, an X-ray image can lead to an inconclusive diagnosis. It will be helpful if one can generate a 3D volume from one or multiple X-ray images.

It would be beneficial to briefly introduce the concept of uncertainty in medical diagnosis and its relevance, providing more context for readers unfamiliar with this concept.

The author(s) assume physician diagnosis follows a normal distribution. Is this the same for both random and systematic misdiagnosis?

The author(s) clipped the original chest X-ray images by 3000 x 3000 pixels and resized them into 256 x 256 pixels before input to the model. Why these values, it there something significant about these values? What happens if I choose different values?

Propose the author(s) re-arrange the model methodology and results section to be more structured. Start with Data, pre-processing (data augmentation), model training, Model evaluation and uncertainty.

The saliency maps in the results are not mentioned in the methodology. Where do they come from, and how and why are there being used? This information should be in the methodology section.

Has the author(s) validated their method with an external dataset?

Minor (but important)

On page 5. The sentence: “Four models are transfer learning using pre-trained convolutional neural networks, and another model is a de novo model trained as an autoencoder from scratch” needs to be rephrased to become more understandable.

The authors should label each figure. For example Figure 4. It will be nice to have Figure 4 a) and b). Same with Figures 5, 6 and 7.

6. PLOS authors have the option to publish the peer review history of their article (what does this mean?). If published, this will include your full peer review and any attached files.

**Do you want your identity to be public for this peer review?** For information about this choice, including consent withdrawal, please see our Privacy Policy.

Reviewer #1: No

Reviewer #2: Yes: Rami Manaf Abdullah

Reviewer #3: Yes: Sandeep Chaurasia

Reviewer #4: No

---

## [Decision Letter · Decision Letter 1]

9 Jan 2024

PDIG-D-23-00283R1

Quantitative evaluation model of variable diagnosis for chest X-ray images using deep learning

PLOS Digital Health

Dear Dr. ONO,

Thank you for submitting your manuscript to PLOS Digital Health. After careful consideration, we feel that it has merit but does not fully meet PLOS Digital Health's publication criteria as it currently stands. Therefore, we invite you to submit a revised version of the manuscript that addresses the points raised during the review process.

Please submit your revised manuscript within 30 days Feb 08 2024 11:59PM. If you will need more time than this to complete your revisions, please reply to this message or contact the journal office at digitalhealth@plos.org. Please include the following items when submitting your revised manuscript:

We look forward to receiving your revised manuscript.

Kind regards,

Ryan S McGinnis

Academic Editor

PLOS Digital Health

Journal Requirements:

Additional Editor Comments (if provided):

Thank you for your resubmission. Please address the final small comments from the remaining reviewer.

Reviewers' comments:

Reviewer's Responses to Questions

**Comments to the Author**

1. If the authors have adequately addressed your comments raised in a previous round of review and you feel that this manuscript is now acceptable for publication, you may indicate that here to bypass the “Comments to the Author” section, enter your conflict of interest statement in the “Confidential to Editor” section, and submit your "Accept" recommendation.

Reviewer #1: All comments have been addressed

Reviewer #2: All comments have been addressed

Reviewer #4: All comments have been addressed

2. Does this manuscript meet PLOS Digital Health’s publication criteria? Is the manuscript technically sound, and do the data support the conclusions? The manuscript must describe methodologically and ethically rigorous research with conclusions that are appropriately drawn based on the data presented.

Reviewer #1: Yes

Reviewer #2: Yes

Reviewer #4: Yes

3. Has the statistical analysis been performed appropriately and rigorously?

Reviewer #1: Yes

Reviewer #2: I don't know

Reviewer #4: Yes

4. Have the authors made all data underlying the findings in their manuscript fully available (please refer to the Data Availability Statement at the start of the manuscript PDF file)?

Reviewer #1: No

Reviewer #2: No

Reviewer #4: Yes

5. Is the manuscript presented in an intelligible fashion and written in standard English?

Reviewer #1: Yes

Reviewer #2: Yes

Reviewer #4: Yes

6. Review Comments to the Author

Reviewer #1: Thanks to the authors for the detailed and organized revision. The aim of this paper is now sharpened and its purpose is clearer. 

Minor:

The GitHub repo is still unavailable. Maybe it’s set to private? I tried to look in the user’s page but there is no mention of this project. 

“One-tenth of the samples (Fig. 2) The Adaptive” missing period after fig 2.

Nitpicking:

“the proposed method was applied using multiple deep learning models with different architectures and a de novo model based on an autoencoder without

transfer learning.” -> “the proposed method was applied using multiple pre-trained and randomly initialised deep learning models with different architectures.”

“1.0 × 10−4” -> “10−4”

I’d suggest shortening the abstract to start from “the purpose of this study..”. The intro before is too long for an abstract but that’s a personal taste.

It’s clear that the legends of Fig. 6 and 7 are duplicated. For the reader’s convenience, I’d consolidated the 12 panels of Figs 6 and 7 into one Fig with 2 panels- PT and S. each panel will show the performance of all the tested models (hued by model)

Reviewer #2: The github link is still broken

Reviewer #4: The authors have addressed my comments

7. PLOS authors have the option to publish the peer review history of their article (what does this mean?). If published, this will include your full peer review and any attached files.

**Do you want your identity to be public for this peer review?** For information about this choice, including consent withdrawal, please see our Privacy Policy. 

Reviewer #1: No

Reviewer #2: Yes: Rami Manaf Abdullah

Reviewer #4: No

---

## [Editor Report · Decision Letter 2]

4 Feb 2024

Quantitative evaluation model of variable diagnosis for chest X-ray images using deep learning

PDIG-D-23-00283R2

Dear Dr. ONO,

We are pleased to inform you that your manuscript 'Quantitative evaluation model of variable diagnosis for chest X-ray images using deep learning' has been provisionally accepted for publication in PLOS Digital Health.

Best regards,

Ryan S McGinnis

Academic Editor

PLOS Digital Health

Thank you for addressing the remaining reviewer comments.